# The Mediating Effect of Fatigue on the Nature Element, Organisational Culture and Task Performance in Central Taiwan

**DOI:** 10.3390/ijerph19148759

**Published:** 2022-07-19

**Authors:** Omar Hamdan Mohammad Alkharabsheh, Amar Hisham Jaaffar, Ying-Chyi Chou, Erni Rawati, Pok Wei Fong

**Affiliations:** 1Department of International Business, Faculty of Accountancy and Management, Universiti Tunku Abdul Rahman (UTAR), Sungai Long 43000, Malaysia; hamdan@utar.edu.my; 2Institute of Energy Policy and Research (IEPRe), Universiti Tenaga Nasional, Kajang 43000, Malaysia; 3Department of Business Administration, Centre for Healing Environment Administration and Research (HEAR), Tunghai University, Taichung City 407224, Taiwan; ycchou@thu.edu.tw (Y.-C.C.); ernirawati94@gmail.com (E.R.); 4Department of Economics, Faculty of Accountancy and Management, Universiti Tunku Abdul Rahman (UTAR), Sungai Long 43000, Malaysia; pokwf@utar.edu.my

**Keywords:** nature element, organisation culture, fatigue, task performance

## Abstract

In the current dynamic business environment, managing the physical working environment of the workforce has become an important part of the company. This study seeks to investigate the effects of the nature element and organisational culture on the task performance of employees with fatigue as a mediator, based on a sample of 103 white-collar employees who work in the central district of Taichung City during the spring and using a purposive sampling method. The data were collected through a self-administered subjective measurement instrument questionnaire and were analysed using Structural Equation Modelling (SEM) path analysis. The results show that organisational culture and task performance were significantly positive. The mediating effect of fatigue caused this relationship to become negative. It was also found that there was a negative relationship between nature elements and task performance. The results provide insights into the importance of employers in providing a healthy workplace which promotes collaboration, health, safety, and the wellbeing of the employee in line with the recommendations of the World Health Organisation (WHO). The study concludes that future international comparative studies can be performed to identify the best workplace design that can reduce employees’ fatigue and alleviate their current work performance.

## 1. Introduction

These days, building design has become a concern, not only from a design or sustainability perspective but also for its impact on human living. Some companies have already been concerned about the green living concept of building offices, houses or apartments. The Industrial Revolution of 1750–1850 brought about significant changes in industry, manufacturing, technology, and transportation which had a great impact on the social, industrial, economic, and cultural conditions of the world [1]. Following the Industrial Revolution, a good percentage of the world’s population migrated to urban areas to live near and work for manufacturing companies. Concerns are now being raised about the indoor environmental qualities of buildings due to the recognition of the hazardous effects of passive smoke and poor ventilation within factories; therefore, the first step is the improvement of the air quality of indoor environments. Many employees are concerned about the internal environmental quality of their offices and workspaces because they carry out most of their tasks inside the office or factory, and a poor internal environment could negatively impact their health due to building-related illnesses. According to Arthur and Powell [2], beyond ventilation, these damaging factors also include ergonomic (lighting, noise, and furniture) and psychological issues (decision latitude, working conditions, and job satisfaction). 

Moreover, much academic research has helped set standards to measure indoor environmental quality (later called IEQ) as a benchmark to design and build homes, apartments, schools, offices, hospitals, health centres, and even gardens and walkways. Additionally, some tools like ASHRAE, HOPE, or the CBE Survey are used to classify the indoor environment quality into eight categories, mostly based on light, noise, thermality, air quality, and layout. Recent research into IEQ relates to work productivity, human comfort, and wellbeing. A high-quality indoor environment is believed to improve and enhance human comfort and wellbeing as well as reduce absenteeism and turnover rates among employees [2].

Over the past decade, European workplace design has transformed into today’s modern open office spaces. However, many researchers found that open offices often create greater disadvantages for workers’ comfort, mood, and performance than cell offices [3]. In an open office space, workers tend to have less personal control over the lighting, temperature, and noise. Furthermore, employees cannot regulate their seat position based on their preferences and behaviour. Some people tend to choose a seat position near the window and like to have a big office window that allows direct sunlight to come in. According to Abboushi et al. [4], window views have a direct impact on physical and psychological comfort.

Physical environments are closely related to human psychological states. People have a positive mood when they are satisfied with the environment around them and can personally regulate their environment according to their desires and moods. Other environmental elements such as natural elements (sunlight exposure and greenery factors) can also affect the individual’s psychological condition [5]. The nature element has an invigorating effect that can improve an individual’s psychological condition because, since ancient times, humans have naturally resonated with nature.

Inter-personal relationships are also among the factors that greatly affect the psychological state of the individual because we know that humans need social interaction to ease stress or depression. The quality of social interactions among employees in a firm can be influenced by the organisation’s culture (also referred to as organisational climate) [6]. Good organisational culture can provide positive support and motivation and enhance relationships among colleagues and superiors. The support and morale derived from good organisational culture can provide a sense of belonging to employees (both colleagues and superiors), which in turn enhances their task performance towards achieving organisational goals [7]. On the other hand, employees burdened with heavy workloads are more prone to mental fatigue and cognitive distractions. However, the stress and mental fatigue caused by heavy workloads can be mitigated by the presence of nature elements and a good organisational culture. The restorative effects of nature elements can help alleviate stress and mental fatigue and enhance the cognitive capacity of employees to focus on the task performance.

This study aims to probe the relationship between nature elements and organisational culture with the task performance of white-collar employees working in the Taichung City, Taiwan during spring. The mediating effect of fatigue has also been tested on both independent variables. In terms of theoretical significance, this study contributes to the area of the physical and psychological wellbeing of the workforce by identifying the mediating role of fatigue on the relationship of nature elements and organisational culture with task performance. From a practical significant point of views, this study provides importance insights into the importance of employers in providing a healthy workplace which promotes collaboration, health, safety, and the wellbeing of the employee in line with the recommendations of the World Health Organisation (WHO). The remainder of the paper is organised as follows. Section 2 contains the literature review. Section 3 outlines the theoretical model and hypothesis development, while Section 4 discusses the methods. Section 5 covers statistical analysis techniques. Section 6 discusses data analysis and discussion. Section 7 presents Convergence Validity (CR and AVE). Section 8 covers Model Fit Comparison using AMOS. Section 9 details the hypotheses testing and discussions. Our conclusions are detailed in Section 10.

## 2. Literature Review

### 2.1. Nature Element

Most previous researchers define the nature element as sunlight, daylight illumination, window views, or greenery inside a building, but An et al. [8] have defined the nature elements from a general view and divided it into two sub-factors: potted plant availability and sunlight exposure. The inadequacy of daylight or sunlight can result in visual discomfort, mental fatigue, and stress, which in turn negatively impacts the task performance of employees [9]. Generally, when the lighting level is around 1000 lx, employees tend to perform better compared with working in a low light level of 200 lx. Lighting quality by itself is not a determinant of work performance, but lighting quality influences visual aspects which in turn influence work performance [10]. 

Sunlight refers to natural illumination from the sun or sky, which produces daylight and illumination in the environment. Sunlight is considered to be the best form of illumination for work purposes based on the ASHRAE standard of lighting levels, i.e., 500 lx. The inability to see clearly leads to visual discomfort, which in turn causes fatigue; conversely, when the intensity of light is just at the right level, it tends to enhance the mood of individuals. Further, when the intensity of the light becomes too dull or too bright, people’s moods tend to deteriorate [11]. Additionally, improper daylight illumination can cause glare.

### 2.2. Organisational Culture

There are seven climate conditions that make up organisational culture; these are autonomy, integration, involvement, supervisor support, training, and welfare [6,12,13]. The findings from previous studies show that the correlation between office layout features and job satisfaction can be mediated by organisational culture. Additionally, it has been shown that the office layout, when supported by a positive organisational culture, can increase employees’ satisfaction with their working environment [14,15,16]. Zadeh et al. [17] analysed the impact of the presence of windows and daylight on the rate of communication and social interaction among nurses. An organisation’s culture can affect the interactions and relations between members of an organisation. Social interaction and the support of superiors and colleagues can boost the mood of employees, thereby reducing their mental fatigue. Previous studies reveal that although the open office space can afford a good opportunity for social interaction, it deprives employees of their privacy. As a result, this lack of privacy can lead to a downturn in employee performance because it is difficult for them to concentrate on their tasks due to the noise in their work environment. In addition, Alkharabsheh, et al. [18] found that for an organisation’s desire to achieve positive organisational performance, the organisation has to place importance on organisational culture and leadership.

### 2.3. Fatigue

The mental health and task performance of individuals can be impacted by sunlight. Smolders and Kort [19] investigated the effects of 1000 vs. 200 lx on the eye based on self-reporting measures, task performance, and physiological arousal after a mental antecedent condition (fatigue vs. control). Smolders and Kort [19] analysed the effects of bright light on mental fatigue, drowsiness, vigour, mood, self-restraint, task performance, and physiological arousal. They measured these in relation to heart rate, heart rate variability, and electrodermal response (skin conductance). Additionally, they employed the subjective evaluation of the lighting condition, the general perception of the environment, self-reported performance tasks, and physiological measurements. The outcome of their investigation revealed that the effects of light that is too bright or too dull are the same on mental condition and task performance [20]. In order to achieve visual comfort and thereby boost one’s mental condition, individuals require a lighting intensity of about 500 lx [20]. As observed by Lamb and Kwok [11], inadequate office Indoor Environment Quality acts as an environmental stressor which can lead to a decrease in self-reported work performance and stated variables (motivation, drowsiness, and distractibility). In addition, Raanaas et al. [21] observed that the presence of plants in work areas can ease fatigue when proofreading. This is because the presence of plants in the indoor office space can attract the attention of individuals, thereby refreshing their attention capacity, which leads to a decrease in fatigue. Bergefurt et al. [22] identified that some indicators of mental health such as concentration and stress have received less attention in the studies related to the physical workplace, particularly among white collar employers. 

## 3. Theoretical Model and Hypothesis Development

In this research, the nature element and organisational culture act as the independent variables, task performance acts as the dependent variable, and the relationship between organisational culture and task performance is mediated by fatigue. The figure below (Figure 1) presents the research model proposed.

### 3.1. Nature Element and Task Performance

The findings from previous studies reveal that nature elements can improve the task performance results of employees by improving employee attention capacity (through their restorative effect) and by reducing working tension. Sunlight can redeem the negative effects of visual discomfort, mental fatigue, or stressors that can affect employees’ work performance [11].

**Hypothesis** **1** **(H1):***The nature element has a significant relationship with task performance*.

### 3.2. Organisational Culture and Task Performance

Organisational culture is an intangible structure of organisations which is difficult to understand for individuals outside of organisations since it is rooted in the values, beliefs, and assumptions held by organisational members [23]. Good organisational culture provides good support to employees from both superiors and colleagues. Additionally, good organisational culture helps to create healthy working relationships that boost the employee’s sense of belonging within the organisation [24]. Employees who work in firms that possess good organisational culture are inclined to perform better at their tasks because they are well versed about their task and their sense of belonging inspires them to perform well for the corporate good of the organisation [25].

**Hypothesis** **2** **(H2):***Organisational culture has a significant relationship with task performance*.

### 3.3. Fatigue and Task Performance

Employees who are mentally fatigued also experience stress and depression. Mental fatigue leads to reduced energy levels, and as such, employees who experience mental fatigue do not accomplish much during the day, which can then have a negative effect on their task performance results [26]. Due to a poor attention capacity and poor cognitive function, mentally fatigued employees are more likely to perform worse at their tasks than mentally healthy employees. Mentally fatigued employees possess lower attention capacities, which results in absentmindedness, a lack of focus during work, and tardiness in completing tasks; therefore, their task performance results suffer [26].

**Hypothesis** **3** **(H3):***Fatigue has a significant relationship with task performance*.

### 3.4. Fatigue Mediation on Nature Elements in Relation to Task Performance

The nature element has been observed to help to ameliorate negative mental states such as fatigue, stress, and depression among employees. By providing a restorative and more relaxed atmosphere, nature elements can lead to positive mental health among employees. A positive and relaxed atmosphere can make employees perform better at their tasks because a positive and relaxed atmosphere helps enhance attention capacity and positive mood [27].

**Hypothesis** **4** **(H4):***Fatigue has a mediating effect on the nature element’s relationship with task performance*.

### 3.5. Fatigue Mediation on Organisational Culture in Relation to Task Performance

Though a firm may possess a good organisational culture, with a heavy workload and a negative working environment, employees could experience more tension, depression, and mental fatigue [25]. Employees need more cognitive function and energy to perform their tasks well. Apart from good organisational culture, employees require other support factors to ease the stress or fatigue that comes as a result of a heavy workload, so they can perform better at their tasks [24,28]. Furthermore, previous studies have shown that organisational culture has a significant influence on employees’ fatigue [29,30].

**Hypothesis** **5** **(H5):***Fatigue has a mediating effect on the organisational culture in relation to task performance*.

## 4. Methods

This study utilised a purposive sampling design. The target population for this research was employees at white-collar jobs who work in office buildings in the central area of Taiwan, and a judgement sampling method was the sampling method engaged in this research. The sample utilised in this research was employees who held different views about job responsibility and workload. This method was chosen to ensure that the sample covered a broad spectrum of working situations of white-collar workers in central Taiwan. 

This study used the Drop-off and Pick-up Method (DOPU) to reduce potential nonresponse bias [31]. The participants were duly informed that this research was a study about the effects of office working environments and nature elements (divided into sunlight exposure and nature exposure) on their mental fatigue, which in turn affects the performance of their tasks. Thereafter, the participants were requested to complete a questionnaire that contained four parts: basic individual factors, nature elements (sunlight exposure (SE) and nature exposure (NE)), organisational culture (OC), fatigue (FT), and task performance (TP).

### 4.1. Measurement of the Nature Element

The nature element is one of the two independent variables in this study. The nature elements’ variable engaged in this study is adapted from the research by An et al. [8], which investigated, from a general point of view, the restorative effects that nature elements have on the mental health (fatigue) and work attitudes of employees. Furthermore, the nature elements were divided into two sub-factors by An et al. [8], and these sub-factors are: exposure to nature elements (such as indoor potted plants, paintings, or photographs) and exposure to sunlight. There were nine questions related to measuring the exposure to nature elements and eight questions employed to measure the exposure to sunlight; both measurements were carried out using a 5-point Likert scale (1 = strongly disagree; 2 = disagree; 3 = neither agree nor disagree; 4 = agree; 5 = strongly agree).

### 4.2. Measurement of Organisational Culture

Organisational culture was the other independent variable employed in this research. In measuring organisational climate, this study adopted the Organisation Climate Questionnaire designed by Elsa et al. [13]. This abridged version of the Organisation Climate Questionnaire was made up of 15 questions (from 50 questions). The items were measured with the aid of a 5-point Likert scale (1 = strongly disagree; 2 = disagree; 3 = neither agree nor disagree; 4 = agree; 5 = strongly agree).

### 4.3. Measurement of Fatigue

This study used mental fatigue as a mediating variable. It has been identified in earlier studies that “workload” includes the organisational culture that can affect the employees’ mental health (fatigue) and that nature elements can offer some restorative effects to enhance the mental health of employees. This research utilised the Fatigue Assessment Scale proposed by Michielsen et al. [32]. This same scale was also utilised by Shaw et al. [33] in their research into individual differences in personality, ability, and states of stress. The Fatigue Assessment Scale consists of ten questions using a 5-point Likert scale (1 = never; 2 = sometimes; 3 = regularly; 4 = often; and 5 = always). To equalise the measurement standards, a 5-point Likert scale was adapted to become: 1 = strongly disagree, 2 = disagree, 3 = neither agree nor disagree, 4 = agree, and 5 = strongly agree.

### 4.4. Measurement of Task Performance

In measuring task performance, this research made use of the task performance dimension of the Individual Work Performance Questionnaire (IWPQ) from Koopmans et al. [34]. A total of 13 questions were contained in the task performance dimension (Koopmans et al. 2014). In the Individual Work Performance Questionnaire (IWPQ) sub-dimensions of task performance, the numbers 1–6 have a different measurement scale, and it asks more about the workload of the participants. In order to equalise the question items utilised in this study, only questions 7–13 were adopted as the study’s questionnaire question items in order to measure the perceived task performance of the participants. The measurement was based on a 5-point Likert-scale, which was adapted to become: 1 = strongly disagree, 2 = disagree, 3 = neither agree nor disagree, 4 = agree, and 5 = strongly agree.

## 5. Statistical Analysis Technique

This study utilised AMOS (Analysis of Moment Structures) as the analytical tool because of its user-friendly graphic interface and because it is an easier way of specifying structural models that also support the BASIC programming interface as an alternative [35]. The indicators are observed variables such as question items in a survey instrument; the indicators for each latent variable must be more than three so as to avoid the AMOS’s failure to identify the latent variables. Each indicator pairs with error estimates, which need to be reliable, and indicators should have pattern coefficients (factor loadings) of 0.51 or higher on their latent factors. The AMOS regression procedures can be used to measure model fit, modification indexes, the coefficient of determination between latent variables, and the significance of the path between latent variables. In addition, the Structural Equation Modeling (SEM) uses confirmatory factor analysis (CFA) to check the reliability and validity of each latent variables’ indicators (the variable’s questions item) that are measured by analysing the goodness-of-fit and composite reliability index. After the indicators of each latent variable passed the confirmatory factor analysis (CFA) test, the next step was to check the path analysis between each latent variable and the goodness-of-fit of the research model.

## 6. Data Analysis and Discussion

### 6.1. Profile of Respondents

This study used a total of 103 respondents. Of the respondents, 46 were male and 57 were female. The age of most of the respondents ranged between 31–45 years (53.4%), while most of the respondents had a work experience of over 11 years (50.5%). Most of the data were collected from workers in the urban city of Taichung; 76.7% of these workers had window views in their office building. In relation to the window position, 30.1% of the workers had a window view on the left side, while 26.2% had a window view on the right side; also, 23.3% had the window view located behind them. Further, with regard to the office floor level, 35% were located on the 9th–10th floor and 33% were located on the 1st–3rd floor. In addition, the number of hours the employees spent working indoors averaged between 15–25 h per week (27.2%) and 26–35 h per week (26.2%). The number of hours the employees spent working outside the office building averaged between 2–4 h per week (34%), 0–1 h per week (28.2%), and above 7 h per week (23.3%). A majority of the respondents spend more of their working hours inside the office building due to the administrative nature of their jobs.

### 6.2. Reliability and Validity of the Results 

In order to test the reliability of each variable’s items, this research utilised AMOS by implementing the confirmatory factor analysis (CFA), using hierarchy regression to test the proposed research model. The model fit for each question was arrived at using chi-square (df), normed chi-square, Goodness of Fit Index GFI, Comparative Fit Index (CFI), and Root Mean Squared Error of Approximation (RMSEA) to measure the acceptability of the model’s fitness to the data. The validity test employed in this study was composite reliability (Cr) and average variance (AVE).

### 6.3. Confirmatory Factor Analysis

#### 6.3.1. Nature Element

There were eight question items for sunlight exposure and, based on the covariance analysis result, questions 2 and 8 had a factor loading of less than 0.51, respectively, and they also possessed a high Modification Index (M.I) with another question. There were nine questions for nature exposure and, based on the covariance analysis results, question number 8 had a factor loading which was less than 0.51 and also possessed a high Modification Index (M.I) with another question. As shown in Table 1, the sunlight exposure and nature exposure variables had good model fit with the normed chi-square (1 < 1.92 < 3), GFI (0.835 < 0.9), CFI (0.927 > 0.8), and RMSEA (0.095 < 0.1).

#### 6.3.2. Organisational Culture

The study questionnaire contained 15 questions for organisational culture. Based on the covariance analysis, there were six questions whose factor loadings were less than 0.51, and they also possessed high Modification Index (M.I) with another question. As shown in Table 2, the organisational culture variable had a good model fit, where the normed chi-square was (1 < 1.568 < 3), and the GFI (0.919 > 0.9), CFI (0.974 > 0.8), and RMSEA (0.075 < 0.08) met the good model standards.

#### 6.3.3. Fatigue

The study questionnaire contained 10 questions for Fatigue, and based on the covariance analysis, there were 2 questions that had their factor loadings below 0.51 and also possessed a high Modification Index (M.I) with another question. As shown in Table 3, the fatigue variable had good model fit, where the normed chi-square (1 < 1.554 < 3), GFI (0.927 > 0.9), CFI (0.971 > 0.8), and RMSEA (0.074 < 0.08) met the good model standards.

#### 6.3.4. Individual Perceived Task Performance

The study questionnaire contained seven questions for task performance, and, based on the covariance analysis, there was one question that had a factor loading below 0.51 and also possessed a high Modification Index (M.I) with another question. As shown in Table 4, the task performance variable had a good model fit, where the normed chi-square (1 < 1.265 < 3), GFI (0.965 > 0.9), CFI (0.991 > 0.8), and RMSEA (0.051 < 0.08) met the good model standards.

## 7. Convergence Validity (CR and AVE)

In ensuring the validity of the item’s questions used to measure the variables, this study utilised the composite reliability (Cr) and average variance extracted (AVE). In this study, there are four independent variables: sunlight exposure (SE), nature exposure (NE), organisational culture (OC), and cognitive failure questionnaire-distractibility (CFQ), with hierarchy regression using one mediation variable, fatigue (FT), and one dependent variable, perceived individual task performance (TP). The table below (Table 5) shows the various loading factors, composite reliability (Cr), and average variance extracted (AVE) for all the variables’ question items.

## 8. Model Fit Comparison AMOS

This study consisted of two independent variables: nature elements (sunlight exposure and nature exposure) and organisational culture. The dependent variable was task performance, with fatigue acting as the mediator variable. The table below (Table 6) shows the model fit of this study’s research model:

We can conclude from Table 6 that all the research models fit the requirements of a good model (*p*-value < 0.05; 0.8 < CFI < 0.9; RMSEA < 0.08). This result concludes that the three research models are acceptable. The path coefficients and the results of the examination on the hypothesised direct effects can be referred to in Figure 2, which outlines the detailed output of the structural equation model used to examine the research hypotheses.

## 9. Hypothesis Testing and Discussions

In order to verify the hypotheses that this study proposed, the AMOS path analysis was adopted to analyse and substantiate the relation of the variables to one another. The outcome of the AMOS path analysis revealed that the nature element (which consists of nature exposure and sunlight exposure as sub-factors) had a significant negative direct effect on task performance (direct effect = −0.719; coefficient of determination = −0.72; regression weight = 0.009). Additionally, the outcome of the AMOS path analysis revealed the significant positive direct effect that organisational culture had on task performance (direct effect = 0.778; coefficient of determination = 0.78; regression weight = 0.016). Moreover, the outcome of the AMOS path analysis also revealed that fatigue had a significant negative direct effect on task performance (direct effect = −0.266; coefficient of determination = −0.27; regression weight = 0.009). Therefore, by virtue of these, hypotheses H1, H2, and H3 were accepted.

This study deals with the effects of fatigue as a mediator in the relationship between nature elements and organisational culture for task performance. Consequently, the mediating effect of fatigue on the relationship between nature elements and task performance is outlined in Hypothesis 4, while Hypothesis 5 outlines the mediating effect of fatigue on the relationship between organisational culture and task performance. Based on the Fatigue Mediation Path Analysis, fatigue has a partial mediation effect on the relationship between nature elements and task performance. The total effect of the mediation relation is −0.186, with a direct effect −0.719 and an indirect effect 0.533. In this relationship, fatigue acts as a partial mediator because, as previously established, there is a significant negative relationship between the nature element and task performance (direct effect = −0.719; coefficient of determination = −0.72; regression weight = 0.009). Additionally, the nature element has a significant negative correlation with fatigue (direct effect = −0.929; coefficient of determination = −0.93; regression weight = 0.002). The intervention of fatigue as a mediation variable causes the link between nature elements and task performance to become significant and positive (indirect effect NEL to TP = 0.533).

Based on the path analysis, fatigue has a full mediation effect on the relationship between organisational culture and task performance. The total effect of the mediation relation is 0.485, with a direct effect of 0.778 and an indirect effect of −0.293. Fatigue produces full mediation effects because, as previously shown, organisational culture has a significant positive link with task performance (direct effect = 0.778; coefficient of determination = 0.78; regression weight = 0.016), although organisational culture has a significant positive link with fatigue (direct effect = 0.510; coefficient of determination = 0.51; regression weight = 0.007). The intervention of fatigue as a mediation variable caused the link between organisational culture and task performance to become a significant negative indirect effect (indirect effect OC to TP −0.293).

Based on the above statistical analysis, we can conclude that the link between the nature element and task performance is significantly negative. Additionally, it was revealed that an increase in the nature element in the workplace could lead to a reduction in task performance among employees. This might be a result of the distractions that the nature elements could cause for employees, which in turn could lead to a downturn in their task performance. However, organisational culture has a significant positive correlation with task performance; this means that a good organisational culture with a warm and positive organisational climate can boost the task performance of employees. This is the case because a good organisational culture, coupled with a warm and positive organisational climate, provides positive psychological support for employees and reduces the mental stress of employees. These positively affect the mood of employees, which in turn leads to better employee task performance when compared to employees who lack a positive organisational culture and climate.

Based on the results of the statistical analysis presented above, the link between fatigue and task performance is a significant negative relationship. Good mental health (unfatigued), which results in more energy and better cognitive function among employees, will lead to a boost in task performance. Conversely, employees who have poor mental health (fatigue) will experience lower energy, concentration, and response, and these will lead to a reduction in their work performance.

Fatigue has a mediation effect on both the nature element and organisational culture. Observations of the results of the AMOS path analysis reveal that the effect of fatigue on the nature element is that of a partial mediator—this might be due to the mentally restorative effect of nature elements. Mental fatigue and work tension can be relieved with the aid of nature elements. Although paying attention to nature elements can be a source of distraction to employees, which in turn reduces their work performance, the restorative effects of nature elements on mental fatigue can help boost the mood of employees, which leads to less tension in their office environment. This positively affects their task performance. Organisational culture does not have a direct relationship with fatigue, but fatigue intervention can cause a positive relationship between organisational culture and task performance to become negative. An organisation could have a good organisational culture and organisational climate, but too heavy of a workload can cause fatigue to set in, which could lead to poor mental health among employees. Employees who work under conditions of bad mental health, even when they are offered positive support, perform worse at their tasks when compared with employees who have a positive mood and good mental health.

## 10. Conclusions

The mediating effect of fatigue on the nature element and organisational culture reveals the importance of mental health as a significant factor that determines task performance results among employees. Working under heavy workloads and working overtime can lead to bad mental health (for example, fatigue, stress), and this, in turn, could lead to reduced task performance among workers; therefore, a good organisational culture and climate by itself may not always lead to a positive relationship with employees’ task performance. If the workload of employees is not compatible with their abilities, it could result in a decline in their mental state and then to a downturn in employee task performance. This underscores the need for organisations to align the workload of employees with their respective abilities. Although nature elements in and of themselves cannot positively enhance the task performance of workers, the intervention of fatigue can alter the link between nature elements and task performance [21]. The link between the nature element and task performance becomes positive due to the restorative effects of the nature element, which help in ameliorating the fatigue and tension that employees experience while working within the office [11].

Due to time constraints, this study could only use subjective measurements to collect the data and conduct the analysis. The other limitation of this study was that the research sample was too small, as only 103 employees were sampled, which is inadequate in terms of the white-collar employees in Taiwan. Initially, this study set out to compare the white-collar employees who work in an office with natural views or urban views from their office windows, but it was difficult to find an office building that had a “natural view” in the heart of Taichung City, as Taichung City is mostly an industrial area with large factories and high office buildings. This research could be improved in the future by including some other indoor environmental factors, such as noise, visual discomfort, air quality, ventilation systems, or even the office’s indoor wall colour, that could affect employees. 

## Figures and Tables

**Figure 1 ijerph-19-08759-f001:**
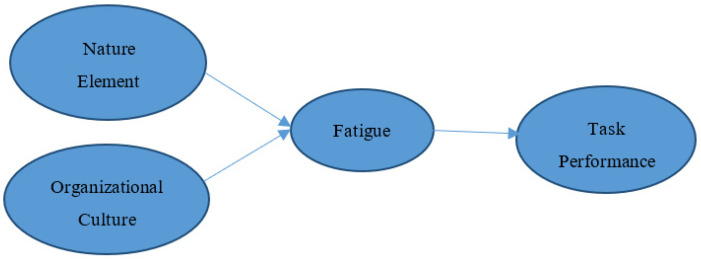
Proposed Research Model.

**Figure 2 ijerph-19-08759-f002:**
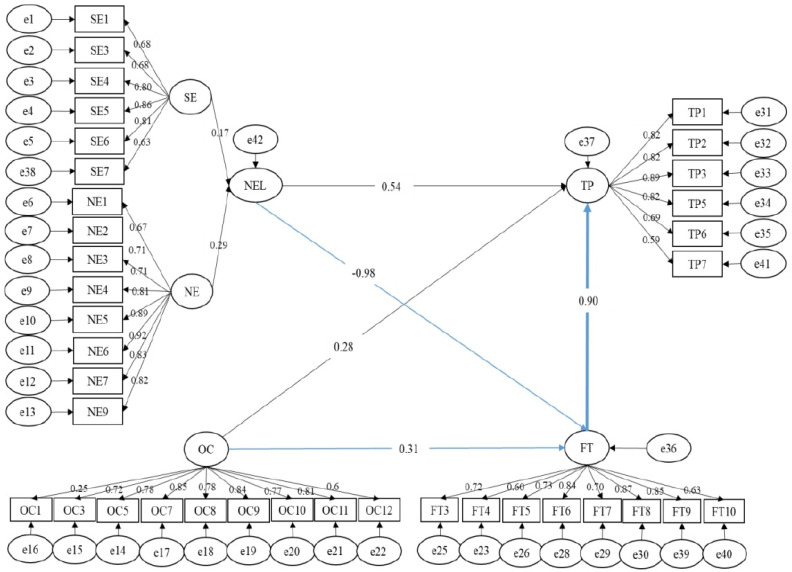
Mediation Model of Fatigue on Nature Elements and Organisational Culture’s Relation towards Task Performance.

**Table 1 ijerph-19-08759-t001:** Nature Element Model Fit.

Chi-Square (df)	Normed Chi-Square	GFI	AGFI	CFI	RMSEA
145.958 (76)	1.92	0.835	0.772	0.927	0.095

**Table 2 ijerph-19-08759-t002:** Organisational Culture Model Fit.

Chi-Square (df)	Normed Chi-Square	GFI	AGFI	CFI	RMSEA
42.324 (27)	1.568	0.919	0.865	0.974	0.075

**Table 3 ijerph-19-08759-t003:** Fatigue Model Fit.

Chi-Square (df)	Normed Chi-Square	GFI	AGFI	CFI	RMSEA
31.077 (20)	1.554	0.927	0.869	0.971	0.074

**Table 4 ijerph-19-08759-t004:** Task Performance Model Fit.

Chi-Square (df)	Normed Chi-Square	GFI	AGFI	CFI	RMSEA
11.383 (9)	1.265	0.965	0.918	0.991	0.051

**Table 5 ijerph-19-08759-t005:** Research Variables’ Question Item Convergence Validity.

Construct	Item	Factor Loading	Average Variance Extracted (AVE) a	Composite Reliability (CR) b
Sunlight Exposure (SE)	SE1	0.69	0.557	0.882
	SE3	0.69		
	SE4	0.79		
	SE5	0.85		
	SE6	0.81		
	SE7	0.62		
Nature Exposure (NE)	NE1	0.66	0.640	0.933
	NE2	0.71		
	NE3	0.71		
	NE4	0.81		
	NE5	0.89		
	NE6	0.92		
	NE7	0.84		
	NE9	0.82		
Organisational Culture (OC)	OC1	0.780	0.600	0.930
	OC3	0.720		
	OC5	0.790		
	OC7	0.860		
	OC8	0.770		
	OC9	0.830		
	OC10	0.790		
	OC11	0.810		
	OC12	0.590		
Fatigue (FT)	FT3	0.650	0.512	0.891
	FT4	0.560		
	FT5	0.700		
	FT6	0.810		
	FT7	0.650		
	FT8	0.850		
	FT9	0.840		
	FT10	0.600		
Task Performance (TP)	TP1	0.760	0.541	0.873
	TP2	0.790		
	TP3	0.880		
	TP5	0.760		
	TP6	0.630		
	TP7	0.540		

a, b: Table 5 shows that all the variables’ questions items are valid (CR > 0.5, AVE > 0.5).

**Table 6 ijerph-19-08759-t006:** Research Model’s AMOS Model Fit.

Models	Normed x^2^ (df)	*p*-Value	GFI	AGFI	CFI	RMSEA
Full Model	1.649 (1250)	0.000	0.650	0.606	0.816	0.060

Notes: N: 103, x^2^ = chi-square discrepancy; df = degree of freedom; CFI = comparative fit index; NFI = normed fit index; RMSEA = root mean square error of approximation. All models are compared to the full measurement model.

## Data Availability

The data presented in this study are available on request from the corresponding author. The data are not publicly available to protect respondent privacy.

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
