# Peer review of "The Mediating Effect of Fatigue on the Nature Element, Organisational Culture and Task Performance in Central Taiwan"

_ijerph, 2022, doi:10.3390/ijerph19148759_

Round 1

Reviewer 1 Report

  1. The composition of the sample is insufficiently argued, correlated to applied instruments of research. Should be improved.
  2. What are the main lessons identified and learned and what is the possible manner to extend the results of such a local study to national level and why not, to the level of existed or new theory in the field? To be developed a such idea.
  3. It must be added annexes in order to give a clear image of the intended and applied structure of survey, and subsequently, of the main results.
  4. It should be clarified in abstract section the relation between nature element and task performance in order to be expressed explicitly, without semantic contrast.

Author Response

Dear respected reviewer. Kindly please find my response to your comment in the attached table. Thank you. 

Reviewer 2 Report

Thank you for a possibility to get to know the insights of the research. After reading the paper questions appear and invites for some additional input into the presentation of the research.

Please see my comments.

General remarks:

  • Abstract is long but simultaneously misses some substantial information such as research question, implications of the research and future research directions.
  • More substantial results should be presented in the abstract since stating that heavy workloads and overtime can lead to bad mental health sounds very obvious without any significant research. More grounded results and conclusions would be useful to present in the abstract.
  • Some important information is missing in the introduction such as the objective and tasks of the research, research question and structure of the paper.

Literature review:

  • Analysis of the findings of the existing studies especially recent research papers that explore individual dimensions (task performance, fatigue, organisational culture, natural indicators) as well as the relationship between them. A more extensive analysis of the scientific research could give a clear understanding of the research gap that is tackled with this research paper.
  • The novelty of the research presented is not revealed. This could be disclosed after the research gap is introduced.
  • Organisational culture according to some scholars is not related to the climate although there are debates. Understanding that different scholars might follow different approaches please expand on this (lines 163-167).
  • The direction of the relationship of the dimension fatigue and the others are not so clear and it is not clearly explained 

Materials and Methods:

  • The determination of the indicators to assess the dimensions (task performance, fatigue, organisational culture, natural indicators) is explored in the paper are missing, e.g., there is no explanation what indicators and what is the scientific background for it are used to assess organisational culture and its relationship with other dimensions.
  • The research protocol and a detailed step-by-step research design should be introduced. The explanation of the methods for data collection, processing and analysis should be introduced and the scientific sources that use the research methods for similar research objectives should be showed. This would serve as a reliability basis of the research.
  • Tables and especially Figure 2 require more analytical work and insights
  • The questionnaire and the primary data are not available. It is not the most important usually but the possibility to get to know the instrument and primary data could demonstrate more clarity and validity of the research presented.

Discussion, conclusions and future research

  • The section Discussion invites to debate own results with the conclusions from other scientific researches and insights.
  • Debates in the section Discussion with the authors cited in the previous sections as well as integration of the findings of their studies could lead to more grounded conclusions.
  • There are no sections for the academic and practical implications and also the directions of the future research which would serve for improvement of the article presented for publication.

Author Response

Dear Respected Reviewer. Kindly please find my response on your comment in the attached document. Thank you. 

Round 2

Reviewer 1 Report

My observations were implemented in a good level of accuracy.

Reviewer 2 Report

The authors made corrections after the review. Addition proofreading could be useful to make reading easier.